# Low contraceptive utilization among young married women is associated with perceived social norms and belief in contraceptive myths in rural Ethiopia

**Tariku Dingeta**[1]*, **Lemessa Oljira**[1], **Alemayehu Worku**[2], **Yemane Berhane**[3]

**1** School of Public Health, College of Health and Medical Sciences, Haramaya University, Harar, Ethiopia,
**2** Department of Epidemiology and Biostatistics, School of Public Health, Addis Ababa University, Addis
Ababa, Ethiopia, **3** Department of Epidemiology, Addis Continental Institute of Public Health, Addis Ababa,
Ethiopia

\* tarikuud@gmail.com

## Abstract

### Introduction

Despite the increasingly wider availability of contraceptives and the high levels of unmet
need for family planning in rural Ethiopia, contraceptive utilization among young married
women is low. Studies on associated factors in Ethiopia so far have been focused on individual factors with little emphasis on socio-cultural factors. This study aimed to assess the
association between contraceptive utilization and socio-cultural factors among young married women in Eastern Ethiopia.

### Methods

A community-based survey was conducted among young married women aged 14–24
years. A total of 3039 women were interviewed by trained data collectors using a structured
questionnaire. Adjusted Odds Ratio (AOR) with 95% Confidence Intervals (CI) was used to
identify factors associated with contraceptive utilization using multivariable logistic regression analysis.

### Results

The current contraceptive prevalence rate was 14.1% (95% CI: 12.8–15.5). Perceived
social approval (AOR = 1.90; 95% CI = 1.60–2.30) and perception of friends' contraceptive
practice (AOR = 1.34; 95% CI = 1.20–1.54) were significantly and positively associated with
contraceptive utilization. On the contrary, increased score of belief in contraceptive myths
was significantly and negatively associated with contraceptive use (AOR = 0.60; 95% CI:
0.49–0.73). Moreover, recent exposure to family planning information (AOR = 1.67; 95% CI:
1.22–2.28), ever-mother (AOR = 9.68; 95% CI: 4.47–20.90), and secondary and above education level (AOR = 1.90; 95% CI: 1.38–2.70) were significantly associated with higher odds
of contraceptive utilization.

org/10.1371/journal.pone.0247484

Salamanca, SPAIN

**Data Availability Statement:** All relevant data are
within the manuscript and its Supporting
Information files.

**Funding:** TD LO HURG-2018-02-04 Haramaya University www.haramaya.edu.et No-The funders had no role in study design, data collection and analysis, decision to publish, or preparation of the manuscript.

**Competing interests:** The authors have declared that no competing interests exist.

## Conclusion

Only about one-in-seven young married women were using contraceptive methods. Socio-cultural factors significantly influence young married women's contraceptive utilization. Interventions to address social norms and pervasive myths and misconceptions could increase the use of contraceptive methods in young married women.

## Introduction

Sub-Saharan Africa has the highest proportion of 10–24 years young population in the world and half of them are girls [1]. In Ethiopia, more than 17% of adolescents (15–19) and 60% of young adult (20–24) girls are in a marital relationship [2]. Despite a modest decline in the last 10 years still more than 35% of young girls in Eastern and Southern Africa are married before their 18th birthday [3]. Young women face very strong pressure to bear children soon after marriage to prove their fertility potential. Giving birth at an early age and in shorter intervals are well-known factors that compromise the well-being of mothers and their children [4, 5]. In Ethiopia, unintended pregnancy among young married women is very high and constitutes the leading cause of preventable mortality and morbidity [6, 7].

Young married women in low and middle-income countries have high unmet need for contraception. Young women in the region also often used short-term methods that are related to higher failure and discontinuation [8, 9]. Moreover, 90% of the adolescent unintended pregnancies occur among those who had an unmet need for contraception [6]. In Sub-Saharan Africa (SSA), only less than one-third of 15–19 years and less than half of 20–24 years of married women who want to wait two or more years are using some kind of modern contraceptive methods [10]. The contraceptive utilization among 15–19 years old married women in Ethiopia was 36.5% compared to 46.4% among those 25–34 years married women [11].

Contraception among young women who face pressures to bear children is influenced by multiple factors [12, 13]. Demographic and economic characteristics including age, educational status, and wealth status were associated with married women's contraceptive utilization [14, 15]. Individual factors such as knowledge of contraceptive methods and self-efficacy were also shown to be associated with women's contraceptive utilization [16–19]. Studies have also reported the influence of socio-environmental factors such as spousal communication, women's household decision-making autonomy, and inaccessibility of contraceptive services [20–22].

Recently, literature increasingly recognizes the strong influence of perceived social norms and beliefs in contraceptive myths on young married women contraceptive utilization [21, 23, 24]. A perceived social norm is commonly shared beliefs regarding people in their social networks' (reference group) approving of contraceptive practice (injunctive norm), or believing that most people in their network are practicing contraception (descriptive norm) [25–27]. Perception of friends, parents, and husbands' approval influences young married women's contraceptive utilization. Mother in-law's pressure to bear children also preclude young women from using contraceptive methods [23, 28, 29]. The perception of contraceptive behavior of friends had also consistent effects on the probability of contraceptive use. Young women who believed that most of their friends are using contraceptives had a higher likelihood of contraceptive use [30, 31]. Moreover, studies have shown the influence on young married women's contraceptive use of belief in myths and misperception or beliefs in rumors not supported by evidence regarding contraceptive methods [24, 30, 32–36].

Despite recent arguments showed the influence of social norms and negative beliefs about contraceptive methods on young women contraceptive utilization [29], contraceptive researches in Ethiopia had been limited to women's demographic, reproductive and institutional factors. Hence, understanding these socio-cognitive influences on young married women contraceptive utilization is critical for improving the well-being of young mothers and their children [37]. Therefore, this study aimed to identify the association of perceived social norms and belief in contraceptive myths with the young married women contraceptive utilization in eastern Ethiopia.

## Methods and materials

### Study setting and period

A community-based survey was conducted in Kersa Demographic and Health Surveillance System (Kersa HDSS) site, Kersa District, Eastern Ethiopia. Kersa HDSS is established by Haramaya University in 2007. The HDSS constitutes twenty-one rural and three urban kebeles (the smallest administration unit in Ethiopia). Six health centers and nineteen health posts provide primary health care services including family planning in the study district. The HDSS involve 195,341 people and more than 26,061 households. More than 80% of married women in the study site were married before 20 years and about 75% of them gave birth before the age of 20 years. These figures are higher compared to the national figure which showed 69% of married reproductive-age women were married before 20 years and 57.4% give birth by the age of 20 years. The total fertility rate (TFR) among reproductive-age women in the study site was 5.5 which is also far higher than 4.6 children per woman (2.3 in urban areas and 5.2 in rural areas) in Ethiopia. The annual population growth rate in the study area was 2.9% [2, 38, 39]. The study was conducted from March 9-May 29, 2018.

### Study population, sampling and sample size

This study population constituted currently married young women, less than 25 years of age, who were permanent residents of the study sites. The Kersa HDSS database was used to identify the list of households with the targeted population. A total of 3102 young ever-married women were identified from the HDSS database. In a community-based survey such as this one, where multiple Likert scale questions are used to measure the exposure variables and there are a large number of covariates, having the largest possible sample size is important to achieve an acceptable margin of error even within the smallest possible subgroup of interest and to adjust for multiple confounding factors. Thus, all the 3102 eligible young married women identified from the KDS-HRC database were included in the study to get possible maximum sample size since the population was well defined (complete sampling frame is available) and small to be interviewed with the manageable resource, time, and quality.

### Data collection

Data were collected through a face-to-face interview by trained interviewers using a structured questionnaire. The English version of the questionnaire was developed by reviewing relevant literature and standard questionnaires. We followed the DHS questionnaire protocol for questions on contraceptive knowledge and utilization. Then, the questionnaire was translated into the local language (Afan Oromo). The questionnaire was intensively reviewed for content and face validity in the context of the study community. The questionnaire was pretested among young married women who were residing in a similar setting outside of the study area. Female data collectors who fluently speak Afan Oromo were recruited and trained on the study tool,

interviewing techniques, and field data collection procedures. Data collectors have interviewed eligible participants who consented to participate through home to home at a private place around their home to keep confidentiality of response. The field research supervisors closely monitored the data collection process and checked the filled questionnaire for completeness and accuracy on a daily basis.

## Ethical considerations

The study was approved by the Institutional Health Research Ethics Review Committee (IHRERC) of Haramaya University, College of Health and Medical Sciences. Written informed consent was obtained from each study participant. The interviewer had read the information sheet to each woman and asked to sign using a fingerprint if the woman couldn't sign by painting colour pen on her finger. The research ethics review committee has waived the consent from parents or guardians of the minors based on article 8.3.5.3 of the Ethiopian national research Ethics review guideline which states the possibility of married or parenting minors to give written informed consent on their own behalf [40].

## Variables and measurements

The outcome variable was current contraceptive utilization. The women were asked if they or their spouse were currently using any contraceptive method to prevent or delay pregnancy. Those who responded 'yes' were further asked which contraceptive methods they were using to identify common method of choices.

Perceived social norms and belief in contraceptive myths were the primary explanatory variables. Perceived social norms include two aspects: Perceived social approval (injunctive norm) and perception of friends' contraceptive practice (descriptive norm). Perceived social approval reflects perceived pressures to conform to others while descriptive norm reflects on the perceived prevalence of the contraceptive utilization [25, 41]. Perceived social approval from people in social network (injunctive norm) was measured by 4 items adapted from literature and modified based on the FGD and IDI conducted before the data collection [42, 43]. The items include "My parents believe I should use birth control to prevent pregnancy, "Most of my friends believe using birth control is important, "My husband believes I should use contraceptives", and "My mother in-law believe I should use contraceptives". The response range from 'strongly disagree' (1) to strongly agree (5) and the index score was constructed by dividing the total scores for the total number of items (S2 Table). The possible score ranges from 1–5 with a higher score indicating that participants perceive their referents approve more their contraceptive use. Factor analysis also showed that all the four items loaded on a single factor with 0.83 Cronbach's alpha, indicating satisfactory reliability. The perceptions of friends' contraceptive practice(descriptive norm) was assessed by a single item adapted from Sutton and Walsh-Buhi [44] asking the respondent to consider her peers (women who are similar to her) and how many were using contraceptives method with the response ranges from 'none' (1) to 'all' (5). The higher the score indicates the stronger the likelihood that respondents thought their peers were using contraceptive methods.

Belief in contraceptive myths was measured based on agreement with 6 specific statements identified from previous literature reflecting contraceptive related myths and misconceptions [45]. The statements include: "Contraceptives are dangerous to women's health, "Contraceptives can harm women's womb", "Contraception would ruin sexual mood", "Use of a contraceptive can make a woman infertile", "Contraceptives can give you deformed babies", and "Women who use contraceptive methods may become promiscuous". Each item was rated on a 5-point scale which ranges from 'strongly disagree' (1) to 'strongly agree' (5) (S1 Table). The

index was constructed by summing the item scores and dividing the total score by the number of items. The possible total score is in the range from 1 to 5, a higher score indicates strong belief in the myths or misconceptions regarding contraceptive methods. Factor analysis showed that all items loaded on a single factor with minimum loading factors of 0.47 and maximum of 0.74 with Cronbach's alpha = 0.75 indicating satisfactory reliability.

The multivariate analysis controls for possible confounding covariates contraceptive self-efficacy, knowledge, inter-spousal communication, household decision-making autonomy, recent exposure to FP information, age, educational status, parity (ever-mother/never-mother), number of desired children, and wealth status as covariates.

Contraceptive self-efficacy was measured by four items asking the respondents how confident or sure they were they could use contraceptives and insist the husband on contraception. The items include how sure was the woman that: she could tell her husband that she wanted to use family planning, . . . she could use family planning, . . . she could discuss topic of family planning with her husband, and . . . she could use family planning even if her husband did not want. The item responses were rated on a 5-point scale ranging from 'not at all sure' (1)" to 'completely sure' (5). The overall contraceptive use self-efficacy score was constructed by summing the item's score and dividing by the number of items [46]. The scale score ranges from 1–5 and a higher score indicates a higher level of confidence to use contraceptive methods. The scale had good reliability (Cronbach's alpha = 0.77).

Spousal communication was measured using 4-items asking the respondent how often the woman may discuss things with her husband [46]. Each woman was asked how often she discusses with her husband things that happened during the day, her worries or feelings, how many children to have, and whether to use contraceptive methods. The item response options range from 'never' (1) to 'always' (4). The overall inter-spousal communication scale was constructed by summing the item's score and dividing by the number of items. The higher spousal communication scale indicates a higher level of inter-spousal communication.

Women's household decision-making autonomy was intended to assess a woman's influence over a range of key household decisions affecting her life and measured using the 10-items scale adapted from literature. The response options for each item included: wife alone, wife and husband together, the husband alone, respondent and another person, and someone else. All indicators converted into binary variables where responses of wife alone or wife and husband together scored as 2 and all other responses scored as 1. The scale score was the sum of item scores divided by the number of items and ranges from 1-2 [46].

Contraceptive knowledge was another covariate assessed using 15 items adapted from literature and emphasizing knowledge of specific contraceptive methods [47, 48] (S2 Table). The responses were scored as "0" for an incorrect answer and "1" for a correct answer and total score obtained by summing all items. The score ranges from 0 to 15 points and dichotomized as good knowledge of contraception if median score and above, and low knowledge of contraception if less than the median score.

Exposure to FP information in the last 12 months was also measured by asking the respondents whether they were told about FP during their visit to a health facility, or by a Health Extension Worker (HEW) or other Health care Workers (HCWs) who visited their home; or heard on radio, TV; or read on newspaper or magazine, posters/leaflets; and/or discussed FP at community event/conversation. The variable was disaggregated into two categories as exposed to FP information if yes to at least one of the question and no otherwise [49].

Wealth index was computed by employing principal component analysis (PCA) based on items assessing household facilities and possessions. Five components were extracted based on eigenvalues >1, factor loadings > |0.3| and the cumulative proportion of variance explained by each component. The score of the first component or factor comprising several heavily

loaded variables and accounting for the largest variation in the data was categorized into quintiles where each individual falls into a poorest, poorer, medium, richer and richest wealth index. Additionally, socio-demographic characteristics included in the analysis are current age, age of marriage, educational status, place of residence (rural and urban), alive and number of desired children.

## Data management and analysis

The completed questionnaires were double entered into EpiData Version 3.1 and analyzed using STATA 14 statistical software. The missing values of each variable were less than one percent, and the chance of missing was unrelated to any of the variables or missing completely at random. Hence, a complete case analysis was used to handle the missing data. Proportions with 95% confidence intervals were calculated for categorical variables and means with standard deviation calculated for continuous variables. Bivariate analysis was carried out between contraceptive use and the main independent variables and covariates. After checking for multicollinearity by examining the correlation matrix, hierarchical multivariate logistic regression was executed to determine the association of main independent variables with contraceptive utilization after adjusting for other covariates. The first model contained only the main independent variables. Due to multicollinearity between age and parity of women, the second and third multivariate regression models were built by excluding the parity and age of women respectively. Akaike's information criterion (AIC) was used to assess the goodness of fit and inform the selection of models. The model with the lowest AIC value is model 2b, including parity and excluding age (Table 2). Odds Ratio (OR) was used to evaluate the association at a p-value ≤0.05 significance level.

## Results

Among the total 3102 young married women identified from the DHSS data base, 3039 were interviewed making the response rate 97.9%. The mean (±SD) age of study participant was 19.6 (±2.6) years. About half, 1487 (48.9%) were married before 18 years with 17.7 (±1.8) years mean (±SD) age at marriage. The majority of the participants were a rural resident, 2805(92.3%), and Muslim, 2934 (96.5%). More than half of the participants 1626(53.5%) has no formal education. The majority of the participants reported being ever pregnant, 2431 (80.0%), 2,214(72.1%) had a minimum of one child, and 435 (14.4%) were pregnant. The median reported walking distance from the nearest health facility was 30 minutes and 72.0% of the respondents reported living within 1 hour walking distance from the nearest health facility (Table 1).

Ever use of contraceptive methods was reported by 695 participants (22.9%, 95% CI: 21.2%-24.6%). Among the study participants, 369 (14.2%, 95% CI: 12.8%-15.5%) were currently using a contraceptive method. The mean (±SD) duration of continuous utilization among currently using contraceptive was 9.4 (±8.9) months with a range of 1–36 months. Current contraceptive utilization was significantly higher among young adults than adolescent women (p-value <0.0001). Injectable was the most used method 172(47.2%) and nearly all 355(96.2%) were using the contraceptive methods to delay rather than limiting pregnancies. Only 17(2%) never mother women were currently using contraceptive methods (p-value <0.001). More than two-thirds 255(69.1%) of contraceptive users were receiving the method from health center (Fig 1).

### Factors associated with contraceptive utilization

In multivariable logistic regression, perceived social approval (AOR = 1.9; 95% CI: 1.60–2.30) and perception of friends' contraceptive practice (AOR = 1.34; 95% CI: 1.20–1.54) were

**Table 1. Background characteristics of married young women in Kersa HDSS, Eastern Ethiopia, 2018.**

| Variables | Response category | Frequency | % |
|---|---|---|---|
| Age of participants in years | 14–17 | 609 | 200 |
| | 18–19 | 628 | 20.7 |
| | 20–24 | 1802 | 59.3 |
| Age of marriage in years | 13–17 | 1487 | 48.9 |
| | 18–19 | 1088 | 35.8 |
| | 20–23 | 464 | 15.3 |
| Religion | Muslim | 2934 | 96.5 |
| | Christian | 105 | 3.5 |
| Level of education | No education | 1626 | 53.5 |
| | Primary (1–4 grade) | 579 | 19.1 |
| | Secondary and higher (5+ grade) | 834 | 27.4 |
| Husband's main occupation | Farmer | 2706 | 89.4 |
| | Others€ | 322 | 10.6 |
| Number of children alive | No children alive One child | 851 | 28.0 |
| | Two children | 1130 | 37.2 |
| | Three and more children | 861 | 28.3 |
| | | 197 | 6.5 |
| ANC follow up for last pregnancy every had (n = 2407) | Yes | 1,305 | 54.2 |
| | No | 1,102 | 45.8 |
| Ever heard of contraceptive method | Yes | 2788 | 91.7 |
| | No | 251 | 8.3 |
| Advised FP during health facility visit in the last 12 months | No | 2,610 | 85.9 |
| | Yes | 429 | 14.1 |
| Received FP advise from HEWs or others HCWs visited home in the last 12 months | No | 2,633 | 87.0 |
| | Yes | 395 | 13.0 |
| Heard FP via radio in the last 12 months | No | 2,589 | 85.45 |
| | Yes | 441 | 14.55 |
| Seen FP information on television in the last 12 months | No | 2,845 | 93.9 |
| | Yes | 184 | 6.1 |
| Read about family planning in a newspaper or magazine in the last 12 months | No | 2975 | 98.2 |
| | Yes | 55 | 1.8 |
| Read about family planning in a pamphlet/ posters/ leaflets in the last 12 months | No | 2,967 | 97.9 |
| | Yes | 64 | 2.1 |
| Heard FP during community conversation/events in the last 12 months | No | 2,810 | 92.7 |
| | Yes | 221 | 7.3 |

**Abbreviations:** ANC, antenatal care; HEWs, Health Extension Workers; HCWs, Health Care Workers; FP, Family planning.

€ = Other category includes employees, daily laborer and merchants.

significantly associated with higher odds of contraceptive utilization. Differently, the higher the score of belief in contraceptive myths was significantly associated with lower likelihood of contraceptive utilization (AOR = 0.60; 95% CI: 0.49–0.73). Moreover, odds of contraceptive utilization were significantly higher among participants with better contraceptive knowledge (AOR = 1.67; 95% CI: 1.22–2.27), recent exposure to FP information (AOR = 1.67; 95% CI: 1.22–2.28), ever-mother (AOR = 9.68; 95% CI: 4.47–20.90), and who attended secondary education and above (AOR = 1.90; 95% CI: 1.38–2.70) (Table 2).

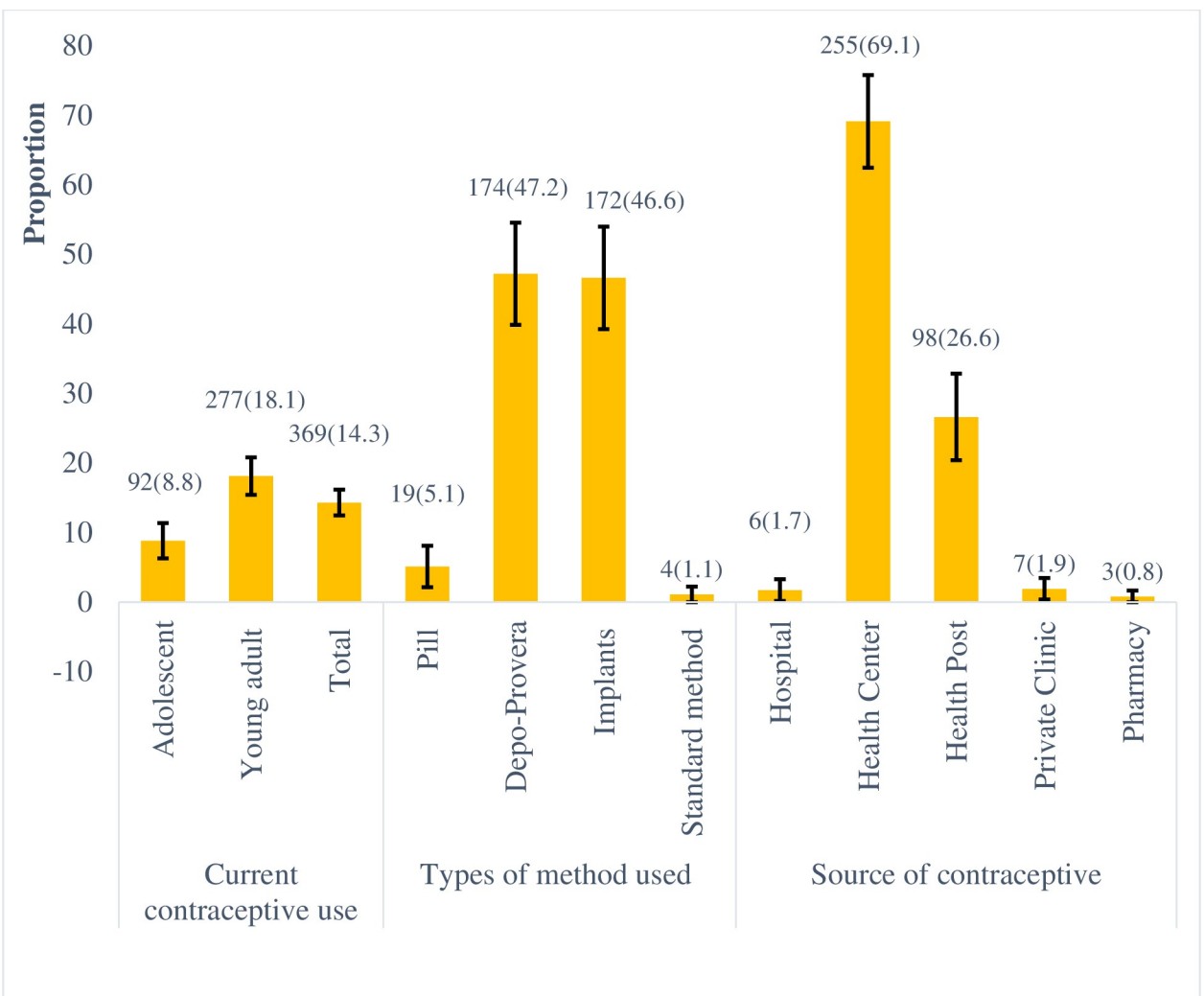

**Fig 1. Current contraceptive utilization of young married women in Kersa HDSS, Eastern Ethiopia.**

## Discussion

This study revealed that less than one in seven young married women were using contraceptive methods. Perceived social approval and perception friends' contraceptive practice were positively associated young women contraceptive utilization. Moreover, the more young women believe in contraceptive myths was negatively associated with their contraceptive utilization.

The current study found that only 14.2% of the participants reported the current utilization of contraceptive methods. This finding is far below the 32–47% level of contraceptive use among 15–24 years married women in a sub-group analysis of studies on reproductive-age women and DHS reports in Ethiopia [11, 50]. Moreover, our finding is far below 40.7% of Oromia, the region where this study was conducted, but higher than 3.4% in the neighboring Somali region [2]. In addition to geographical closeness, the population has close cultural and religious similarities with the Somali region's population, a region with a contraceptive prevalence among the lowest in Ethiopia. Hence, it could be concluded that the prevalence of contraceptive use in Eastern Ethiopia is low. The persistent low contraceptive utilization in the

**Table 2. Bivariate and multivariate logistic regression analysis to show association of young married women contraceptive utilization with perceived social norm and belief in contraceptive myths in Kersa HDSS, Eastern Ethiopia, 2018.**

| Variables | Frequency(%) of contraception | | Crude OR (95% CI) | Adjusted OR (95% CI) | | |
|---|---|---|---|---|---|---|
| | Yes | No | | Model 1 | Model 2a | Model 2b |
| Perceived social approval (injunctive norm) | Mean = 3.3 | Mean = 2.4 | 2.4(2.13–2.74) | 2.1(1.70–2.33)*** | 2.01(1.64–2.40)*** | 1.9(1.60–2.30)*** |
| Perception of friends' contraceptive practice(descriptive norm) | Mean = 3.4 | Mean = 2.5 | 1.8(1.65–2.02) | 1.5(1.30–1.63)*** | 1.41(1.21–1.63)*** | 1.34(1.20–1.54)*** |
| Belief in Contraceptive myths | Mean = 2.1 | Mean = 2.5 | 0.55(0.47–0.65) | 0.58(0.5–0.68)*** | 0.56(0.46–0.69)*** | 0.60(0.49–0.73)*** |
| Age of women in years: 14–17 | 26(5.21) | 473(94.8) | 1 | | 1 | |
| 18–19 | 65(11.90) | 482(88.12) | 2.45(1.53–3.93) | | 1.51(0.84–2.72) | |
| 20–24 | 277(18.10) | 1254(81.91) | 4.02(2.65–6.10) | | 2.11(1.21–3.40)*** | |
| Educational status of women | | | | | | |
| No education | 129(9.5) | 1,233(90.5) | 1 | | 1 | 1 |
| Primary | 94(19.0) | 401(81.01) | 2.2(1.7–2.9) | | 1.50(1.0–2.30) * | 1.70(1.10–2.50) * |
| Secondary and higher | 135(19.8) | 547(80.21) | 2.2(1.72–2.9) | | 1.90(1.33–2.71)*** | 1.93(1.38–2.70)*** |
| Residence: Rural | 298(12.5) | 2,080(87.5) | 1 | | 1 | 1 |
| Urban | 71(36.04) | 126(64.0) | 3.9(2.8–5.4) | | 1.20(0.60–2.40) | 1.12(0.57–2.20) |
| Wealth Index: Poorest | 46(10.20) | 405(89.80) | 1 | | | 1 |
| Poorer | 56(12.50) | 392(87.50) | 1.26(0.83–1.90) | | 11.34(84–2.20) | 1.26(0.78–2.04) |
| Middle | 65(14.04) | 398(85.96) | 1.43(0.96–2.15) | | 0.97(0.60–1.60) | 0.90(0.54–1.44) |
| Richer | 58(12.80) | 395(87.20) | 1.29(0.86–1.95) | | 1.37(0.86–2.17) | 1.22(0.76–1.94) |
| Richest | 62(13.54) | 396(86.46) | 1.38(0.92–2.07) | | 1.01(0.62–1.63) | 0.99(0.61–1.64) |
| Number of desired children: > 5 children | 133(11.34) | 1040(88.7) | 1 | | 1 | 1 |
| ≤ 5 children | 216(18.8) | 933(81.2) | 1.8(1.43–2.3) | | 1.38(1.02–1.85) | 1.35(1.01–1.81) |
| Self-efficacy for contraception | Mean = 3.8 | Mean = 3.2 | 2.0(1.75–2.3) | | 1.1(0.90–1.34) | 1.05(0.87–1.27) |
| Women's household decision making autonomy | Mean = 3.3 | Mean = 2.4 | 2.56(1.56–4.2) | | 1.82(0.90–3.73) | 2.15(1.09–4.22) * |
| Spousal communication | Mean = 1.6 | Mean = 1.8 | 2.0(1.7–2.3) | | 1.10(0.85–1.34) | 1.14(0.91–1.42) |
| Knowledge of family planning | | | | | | |
| High knowledge | 239(18.6) | 1,118(90.53) | 2.2(1.7–2.8) | | 1.67(1.21–2.35)** | 1.67(1.22–2.27) ** |
| Low knowledge | 117(9.5) | 1,044(81.4) | 1 | | 1 | 1 |
| Exposure to FP information in the last 12 months: | | | | | | |
| Yes | 180(23.8) | 1,621(89.8) | 2.7(2.2–3.4) | | 1.54(1.11–2.15)** | 1.67(1.22–2.28)** |
| No | 185(10.24) | 577(76.2) | 1 | | 1 | 1 |
| Parity: Never mother | 17(2.0) | 840(98.0) | 9.6(5.85–15.07) | | | 9.68(4.47–20.90)*** |
| Ever-mother | 351(16.25) | 1809(83.75) | 1 | | | 1 |
| AIC value | | | | 1881 | 1339 | 1281 |

**Abbreviations:** OR, Odds ratio; CI, Confidence interval; FP, Family planning

*p<0.05

** p<0.01

***p<0.001

**Model 1:** Multivariate logistic regression analysis with main independent variables

**Model 2a:** Multivariate logistic regression analysis with main independent variables and all other covariates excluding ever-mother status due to its multicollinearity with age

**Model 2b:** Multivariate logistic regression analysis with main independent variables and all other covariates excluding age due to its multicollinearity with ever-mother status.

area could be due to the socio-cultural pressure to prove fertility immediately after marriage and norms that encourage large family size in rural settings of some low and middle-income countries [36, 51]. This is possibly due to the social pressure on married young women to prove their fertility or have a first child as early as possible after marriage regardless of age. Thus, the finding reinforces the importance of interventions to empower girls and discourage the norm of early marriage to delay the age at marriage among girls in the study area [52]. Lack of negotiation and communication ability with husbands and family members and mistrust towards contraceptive methods might also hinder young women's use of contraceptives [53]. Moreover, health workers in some rural low-income countries are also unwilling to provide advice on contraceptives contrary to their official mandate because of the powerful norms that disapprove of young mother contraceptive use [29, 36].

This study identified significantly higher odds of current contraceptive utilization among young women with higher perceived social approval and friends' contraceptive practice. In line with results from our study, previous studies showed the influences of perceived social support on young married women contraceptive use [18, 54]. Several previous qualitative studies have also recognized the effect of social network approval on young women's contraception. Young women's contraceptive decision is more likely influenced by husbands' expectation of children at younger age and mothers-in-law expectation of grand children [33, 36, 55]. These underscore the persistent influence of external pressures on young married women's contraceptive decisions mainly from husbands and mother in-law. Furthermore, evidence also showed young women's strong willingness of conforming to the social expectation of childbearing [56]. Consequently, the low contraceptive use in our study could be due to many of them complying with the social expectation encouraging childbearing and disapproving of contraception among young women. Young women's abiding of the social expectation may be due to mistrust of contraceptive methods and low skill of negotiation and communication with husbands and other family members. Thus, people in young women's social network mainly husbands and mother-in-law needs to be part of the contraceptive program in a way that builds safe spaces to discuss and learn about contraceptive services.

Moreover, in support of previous evidence, our study established the positive influence of perception of friends' contraceptive practice on young women's contraceptive use [30]. This finding suggested that women can learn or receive support from friends regarding contraception. The more belief in friends' contraception approval would also create a chance of discussing contraception among themselves which helps as a way of gaining knowledge and sharing experiences [53]. However, the negative effect of belief in friends' contraceptive approval may be more pronounced in low prevalence settings like our study area since many may refrain from contraceptive use by wrongly justifying their non-utilization as within the bounds of common practice.

Another important finding was the significant negative association of belief in contraceptive myths with current contraceptive utilization. The more young women believe in contraceptive related myths, the lower likelihood of reporting contraceptive use. This result confirms the previous evidence that identified a negative effect of contraceptive related misconceptions on women contraceptive utilization [35, 45]. Young women in rural Ethiopia enter into marriage at an early age with limited FP information. Thus, they often depend on informal sources of contraceptive information such as friends through which false rumors and contraceptive methods' negative effects are more distributed [29]. Another possible explanation is that some of the contraceptive related myths may be social misconceptions beyond individual beliefs; that leads to social disapproval of contraception during young age. Thus, our findings suggest the need for intervention programs that targeted the identified myths and misconceptions to improve young women's contraceptive use alongside preventing early marriage.

This study relied on self-reported data through face-to-face interviews on culturally sensitive topics. Thus, social desirability biases cannot be ruled out and could result in under-reporting of contraceptive use. Some might also agree or disagree in a socially desirable way with the statements used to assess perceived norms and contraceptive myths. However, we made efforts by ensuring the privacy of respondents during the interview and matching interviewer and respondent by age and sex to minimize the bias. Despite the necessary precaution through improving the clarity of survey items and balancing positively and negatively worded items, respondents may opt for a neutral choice of Likert-scale item or may provide affirmative responses. The positive association between contraceptive use and subjective norms also may be due to contraceptive users' higher likelihood of feeling others' approval than its actual influence on contraceptive use. The traditional norm of not using contraception might perpetuate reporting of the myths that learned from their social networks as barriers to contraceptive utilization. Hence, the relationship of contraceptive myths might be due to post-hoc rationalization of a refusal to contraceptive utilization. Lastly, the social norm is a stem of individual perception of social approval of desired behavior (i.e., perceived social approval), beliefs of what would happen if failing to comply with the approved norm (i.e., possible sanctions), and attitude to comply with the norm (i.e., sensitivity to conform) [57] yet the current study limited to perceived social approval like some other previous studies [18, 54]. Hence, a longitudinal study is needed to develop a full picture of the social norm, understanding the relations between the various constellations, and design tailored intervention to shift the conservative norms which disapprove of young women's contraception.

## Conclusions

Our findings indicate low contraceptive utilization among young married women in rural Ethiopia. Socio-cultural factors, mainly perceived social approval of contraception and belief in contraception myths, were significantly associated with young women contraceptive utilization. Family planning education should address social norms that disapprove of contraception and target myths and misconceptions regarding modern contraceptive methods. Additionally, training of health workers is needed to interact and share information with mothers seeking maternal and child health to dispel pervasive myths and misconceptions.

## Supporting information

**S1 Table. Frequency of each item used for measuring perceived social approval and belief in contraceptive myths among young married women in Kersa HDSS, Eastern Ethiopia, 2018.**
(DOCX)

**S2 Table. Frequency of each item used to assess knowledge of contraceptive methods among young married women in Kersa HDSS, Eastern Ethiopia, 2018.**
(DOCX)

**S1 Data. Minimal data set.**
(DTA)

## Acknowledgments

We would like to thank Haramaya University and Addis Continental Institute of Public health for technical support; the study participants and data collectors for their kind cooperation.

## Author Contributions

**Conceptualization:** Tariku Dingeta, Yemane Berhane.

**Data curation:** Tariku Dingeta.

**Formal analysis:** Tariku Dingeta, Alemayehu Worku, Yemane Berhane.

**Funding acquisition:** Lemessa Oljira.

**Investigation:** Tariku Dingeta, Lemessa Oljira, Yemane Berhane.

**Methodology:** Tariku Dingeta, Lemessa Oljira, Alemayehu Worku, Yemane Berhane.

**Project administration:** Tariku Dingeta.

**Software:** Tariku Dingeta.

**Supervision:** Lemessa Oljira, Alemayehu Worku, Yemane Berhane.

**Validation:** Lemessa Oljira, Alemayehu Worku, Yemane Berhane.

**Writing – original draft:** Tariku Dingeta, Lemessa Oljira, Yemane Berhane.

**Writing – review & editing:** Tariku Dingeta, Lemessa Oljira, Alemayehu Worku, Yemane Berhane.

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
