## [Decision Letter · Decision Letter 0]

15 Jun 2020

PONE-D-20-05351

Low contraceptive utilization among young married women is associated with perceived social norms and believes in contraceptive myths in rural Ethiopia

PLOS ONE

Dear Dr. Dingeta,

Thank you for submitting your manuscript to PLOS ONE. After careful consideration, we feel that it has merit but does not fully meet PLOS ONE’s publication criteria as it currently stands. Therefore, we invite you to submit a revised version of the manuscript that addresses the points raised during the review process.

In particular, reviewers 1 and 3 provide very helpful comments to improve the paper. They agree that the setup needs improvement, in particular posing better the research question. You willl benefit from looking at the strobe statement that provides a checkup of research and reporting soundness, https://www.strobe-statement.org/fileadmin/Strobe/uploads/checklists/STROBE_checklist_v4_cross-sectional.pdf

In addition, as reviewer 3 points out, stating that the data is available without restriction is not enough to meet PLOS ONE data sharing requirements. The data should indeed be available. Check the journal policy, you have a range of options: https://journals.plos.org/plosone/s/data-availability. In your resubmission we invite you to opt for one of them. Otherwise, we would be unable to accept the paper.

We look forward to receiving your revised manuscript.

Kind regards,

José Antonio Ortega, Ph.D.

Academic Editor

PLOS ONE

Journal Requirements:

2. Please include additional information regarding the survey or questionnaire used in the study and ensure that you have provided sufficient details that others could replicate the analyses.

For instance, if you developed a questionnaire as part of this study and it is not under a copyright more restrictive than CC-BY, please include a copy, in both the original language and English, as Supporting Information.

In addition, please provide further details of the pre-testing of this questionnaire, including the exact number of participants and where they were recruited from.

3. You indicated that you had ethical approval for your study.

In your Methods section, please ensure you have also stated whether you obtained consent from parents or guardians of the minors included in the study or whether the research ethics committee or IRB specifically waived the need for their consent.

4. During your revisions, please note that a simple title correction is required to ensure there are no errors of grammar: "Low contraceptive utilization among young married women is associated with perceived social norms and believing in contraceptive myths in rural Ethiopia".

Please ensure this is updated in the manuscript file and the online submission information.

Reviewers' comments:

Reviewer's Responses to Questions

**Comments to the Author**

1. Is the manuscript technically sound, and do the data support the conclusions?

Reviewer #1: Partly

Reviewer #2: Yes

Reviewer #3: Partly

2. Has the statistical analysis been performed appropriately and rigorously? 

Reviewer #1: Yes

Reviewer #2: Yes

Reviewer #3: Yes

3. Have the authors made all data underlying the findings in their manuscript fully available?

Reviewer #1: Yes

Reviewer #2: Yes

Reviewer #3: No

4. Is the manuscript presented in an intelligible fashion and written in standard English?

Reviewer #1: No

Reviewer #2: Yes

Reviewer #3: No

5. Review Comments to the Author

Reviewer #1: Thank you for the opportunity to review this paper. The authors examined the influence of perceived social norms and beliefs about contraception on contraceptive use among married adolescents and young women in Ethiopia. Data is obtained from a community survey of 3039 young married women aged 14-24 years. A multivariate logistic regression analysis was used to identify factors associated with contraceptive use.

General comment:

The topic is a very interesting one that is relevant to family planning policies and programmes. There is a growing body of evidence on the link between perceived social norms/contraceptive beliefs and contraceptive use. Although there is merit in the contribution to knowledge, the paper has some conceptual flaws in the analysis and presentation of the results that should be fixed before consideration for publication. Authors included in their analysis of the effects of beliefs on contraceptive use, participants who are not aware of any contraceptive method. This introduces biases in the results. It is not possible to evaluate someone's beliefs and use of contraceptives when they are not aware of it. Analysis of the link between beliefs and contraceptive use should be restricted to those who have ever heard of contraception. It is not clear what study design was used. The study cannot be generalized given its weak design. There are a lot of grammatical errors that should be corrected by a language expert for it to meet publication standards.

Specific comments

Title:

The title of the paper should be written in a proper way highlighting its core content. Currently, it sounds more like a conclusion of the study rather than a title. It should be revised!. The title includes the text‘…in rural Ethiopia’ yet the sample includes participants from urban settings.

Abstract

There is a lack of clarity on how the association between outcome and independent variables are reported. Report only factors associated with contraceptive use.

Introduction

Outlining a conceptual model would be useful. The authors should also highlight key hypotheses to be tested in the study.

 

Methods:

It is not clear why the study computed a sample size of 1566, yet all 3102 young women in the study area were interviewed. It does not add value to calculate the sample size when a census is used. There need to clarify why all young women were interviewed instead of the number that was sampled.

The inclusion of place of residence, rural vs. urban, is problematic given that the urban sample size is negligible. The result is greatly biased given that the sample was not stratified to allow for comparison between rural and urban.

There is a high correlation between age and number of children that might affect the results when these two variables are included in the regression model. Also, adolescents <19 years are more vulnerable and are not as exposed compared to young women 20-24. The study can be strengthened by comparing participants <19 years with those aged 20-24 years

Result

Table 2: Instead of bi-variate logistic regression, do a simple comparison between contraceptive users and non-users by perceived social approval and beliefs using column totals. It also important to compare the mean scores of index indicators between users and non-users

Table 3: The title is ‘Multivariate logistic regression…’ However, there are some bi-variate results in columns 2&3. These should move to Table 2.

It is not clear which variables are controlled in the multivariate model. This makes it very difficult to infer the effects of perceived social norms and beliefs on contraceptive use. A more precise way to handle this would to running a separate model with the main independent variables without controls and another with the controls.

Review how the logistic regression results are reported. There seems to a confusion between the outcome and independent variables in the way the odds are reported. It is, therefore, very difficult to understand the magnitude and the direction of the effect of the independent factors on the outcome.

Reviewer #2: Comments for manuscript

Low contraceptive utilization among young married women is associated with perceived social norms and believes in contraceptive myths in rural Ethiopia

I believe it is well articulated and useful manuscript. My comments are, in the result section mean and standard deviations should be reported rather than SE.

Discussion: there is low contraceptive utilization in your study area even compared to the surrounding areas. You tried to tell us the presence of persistent socio-cultural norms that encourages large family size and then non-use of contraceptives methods at young age in rural setting. But the utilization is still lower than other rural areas. I suggest to find out more on the possible factors that contributed low utilization in your study area.

Reviewer #3: 1. Introduction: In the final statement of the introduction, in which you describe the aim of the study, it would be beneficial to describe this in more detail, so that you are very clearly stating the aim and the scope of the paper. Stating that the aim is to assess contraceptive behavior is perhaps too broad. Please also see below comments on pieces that could be elaborated on in the introduction, specifically the main independent variables.

2. Study setting: The description of the study site is very useful. It would be helpful to list the statistics you listed for the site for Ethiopia in general, so we can compare it to the average for the country. This will help to put your findings in context.

3. Methods and materials: It would be helpful to clarify how the sample was actually chosen. This section on sampling would benefit from the actual equation used with the inputs provided (contraceptive utilization, precision, design effect, and non-response). Why were all women included if the sample size was far exceeded? Were women contacted in their homes (was this a household survey?) Additionally, since half of the sample had no education, it may be useful to clarify how written informed consent was obtained for those who may not be able to read/write.

4. Variables and measurements: It would be very helpful to understand whether these injunctive norms and descriptive norms fit into a conceptual framework of some sort. How are these norms theorized to affect behavior? It would be very helpful to have a description of these norms/their expected effect on behavior in the introduction section of this paper, to help frame the research you conducted to the audience. Please also specify what all of the response options are for each of the norm and myths and misconception variables.

Did you hypothesize any sort of relationship between your two main predictor variables? It would be helpful to explain how you think the norms as well as the belief in myths/misconceptions might be related, in the introduction. Did you predict that one would be more important than the other, or that while they might be independently associated with the outcome, one would no longer be significant in multivariate analyses?

There are a lot of scale variables included, especially for variables that are not main predictors for the model. You may consider reassessing whether all of these are needed. Additionally, if you decide to keep them all, please elaborate on the items included in the contraceptive knowledge scale.

5. Data analysis: Is table 2 necessary for helping to understand the paper? I didn’t understand the purpose of this analysis from the description in the methods, perhaps because you had already described how the scales were calculated in the previous section of the methods. The Cruide ORs Table 2 may be too much information, and could be considered for an appendix if you decide it is needed. If it is needed, please elaborate on its purpose more in this section of the methods. I also don’t think you need to refer to the table numbers in the methods section.

The sentences where you describe that the scale variables were all included as continuous variables in the model is confusing because you describe how these variables were developed/coded in the variables and measurement section. I think you could put these sentences in the variables section instead.

The data analysis section should also clearly state that multivariate logistic regression was conducted, I don’t think this statement was included.

6. Results: The frequencies for age in Table 1 do not add up to 100. If there are missing data, please clarify this. On the other hand, the frequencies add up to more than your total sample size. Please clarify.

For the duration of continuous utilization, is that among current users only?

I don’t understand the purpose of the Cruide ORs from Table 2 in the body of the paper, perhaps it could be included in an appendix.

7. Results: The text describing Table 3 needs to be heavily revised. The odds ratios are not described in the correct way – the odds are of current contraceptive use, not of the background characteristics. Please revise here and in the abstract and discussion as well. Please also continue to be very clear about the outcome and include “current” in contraceptive use.

It would also be helpful to show the reference groups for all variables, even those with 2 categories. Then you could also add the Frequency(%) of contraception for each variable. I would also put the main predictor variables at either the beginning or the end of the table so they stand out.

8. I think the discussion clearly explores a lot of the main findings from the results. In the paragraph on descriptive norms, in the second sentence, it’s unclear whether the authors are arguing that it’s a misperception that most young women aren’t using contraception, when this study showed only 14% were using. Please elaborate on this possible reason women aren’t using. As I mentioned above, it would be helpful to elaborate on the relationships between these two norms as well as the relationship between norms and believing in myths. Please also spend a bit more time elaborating on suggested interventions and future research, perhaps among positive deviants.

I’m not sure I understand this sentence in the limitations: “Additionally, it could be argued that the positive association between contraceptive use and subjective norm may be due to contraceptives users may more likely to perceive others’ approval than who were not using.” Please elaborate.

Minor/Other:

Was there a minimum age for inclusion in this study?

Figure 1 is very hard to read.

I think this study would strongly benefit from a copy-edit for clarity, grammar, and minor issues with formatting in the text (this, along with the description of the ORs as mentioned above is the reason I selected No for Question 4).

PLOS One author guidelines state that contacting the corresponding author for data is not sufficient to meet their data availability guidelines.

6. PLOS authors have the option to publish the peer review history of their article (what does this mean?). If published, this will include your full peer review and any attached files.

Reviewer #1: Yes: George Odwe. Ph.D.

Reviewer #2: No

Reviewer #3: No

---

## [Author Response · Author response to Decision Letter 0]

10 Jul 2020

attached as "Response to Reviewers"

---

## [Decision Letter · Decision Letter 1]

2 Sep 2020

PONE-D-20-05351R1

Low contraceptive utilization among young married women is associated with perceived social norms and believing in contraceptive myths in eastern Ethiopia

PLOS ONE

Dear Dr. Dingeta,

Thank you for submitting your manuscript to PLOS ONE. After careful consideration, we feel that it has merit but does not fully meet PLOS ONE’s publication criteria as it currently stands. Therefore, we invite you to submit a revised version of the manuscript that addresses the points raised during the review process.

I believe the manuscript has greatly benefitted from most changes. One of the referees agrees with the changes, referee 3 has not answered but I believe that most of the comments have been taken. Making the data available is also a great move that can help increase the visibility of the article.

One of the characteristics that set apart this analysis from others is the focus on young AND married women. There should be some reference on pressure to procreate once married (i.e: https://doi.org/10.20897/femenc/5918, https://doi.org/10.1017/S0021932000003552). When this is understood, the low contraceptive prevalence of this group is understood better: they want to have children, particularly their first child. Probably the “policy” recommendation would not be so much “use contraceptives after getting married” but “marry later” given this cultural association. This should be highlighted to understand the context of such low contraceptive use as commented by reviewer 3.Also, in that respect, it would help to give comparable statistics on contraceptive prevalence for a similar group in Ethiopia from DHS. [I have seen you do it in the discussion comparing with DHS2016 in l. 329-330, I still give the data I looked for MiniDHS2019] For instance, it can be computed from the MiniDHS2019 (https://dhsprogram.com/pubs/pdf/PR120/PR120.pdf). It shows 36.5 prevalence for 474 currently married women 15-19 and 52.2 prevalence for 951 married women 20-24. Based on this a proportion for married women 15-24 can be estimated and provided as a reference. Indeed CP is very low in this region. While DHS does not provide a figure for Kersa, CP is not low in Oromia (40.7 compared to 41.4 nationally). However, neighbouring Ethiopia Somali shows only has 3.4 CP. Is there some clear a priori reason for this low uptake?Differences could also arise due to a different protocol in asking about contraceptive use. This should be acknowledged somewhere as a limitation since the DHS protocol is the standard for measuring CP internationally. Difference might lie in that DHS, before asking about use, asks about knowledge of each method.In contrast, line 333-334 is too vague: “the persistent socio-cultural norms that encourages large family size and then non-use of contraceptives methods at young age in rural setting of developing country”. There are many rural setting in developing countries where it is not the case. Also phrasing in singular form is not correct. Maybe “the persistent socio-cultural norms that encourage large family size and then non-use of contraceptive methods at young ages in this rural setting”There is no clear information on when the survey was carried out. That should be made clear.I also see missing data in the dataset. I guess most correspond to women that are not married. How handling of missing data was carried out should be mentioned in the methods section.I understand your argument of multicollinearity of age and parity, also raised by referee 1, but for the reason given above (pressure to become mother), it would be good to include at least a variable distinguishing ever-mothers. One would expect the variables of age lose importance after including it, but it is ok. Also much lower CP for non-mothers would be expected. Please include the variable in the multivariate analysis since non-mothers are less likely to use contraception. Note that l.205 says you are controlling by parity which seems not to be the case (after the addition of the variable, it will).The title has been changed but it still does not read correctly. “believing” should be “belief”, and it still reads too long. I would suggest “Low contraceptive use, perceived social norms and belief in contraceptive myths among young married women in Eastern Ethiopia”.There is no need to put all results in the abstract, just the main ones, in lines 45-53.The text would still requires copy-editing. Some suggestions in that respect, focusing in the abstract, but there is a need to check everything, particularly the discussion.l. 39-40: “aged 14-24” instead of in the age-group 14-24.L. 45: Remove “the” in the “the perceived social approval”L. 56: Remove “the” in “the sociocultural…”.L. 57: Replace “are necessary” which does not stem from the analysis, and “to improve”, which assumes a value judgement regarding contraceptive use, with “could increase” or something similar.Avoid “unmet needs” (l. 73), use “unmet need”.L. 336: suppress “norm”L. 339-40: Rephrase. “Despite” does not make sense. Also “imperative” is too strong.L. 364 should be “mothers”L. 366: suppress “be”L. 367-68 do not make sense. Rephrase.L. 375: Again the problem with “belief” (noun) and “believe” (verb) but the other way round. It should be “believe in” instead of “belief the”.L. 379: Again rephrase “developing countries”. This example is not representative. Just suppress “in developing countries”. Also “which even accompanied by” is wrong. In general this paragraph needs copy-editing.L- 388 Also check all this page for grammar. Example: “This tended to resulted” maybe “result”. You can use “Grammarly.com” or a similar tool that will indicate all these problems.L. 410: Socio-cultural, not social-cultural. Also commas missing before mainly and after myths.

We look forward to receiving your revised manuscript.

Kind regards,

José Antonio Ortega, Ph.D.

Academic Editor

PLOS ONE

Reviewers' comments:

Reviewer's Responses to Questions

**Comments to the Author**

1. If the authors have adequately addressed your comments raised in a previous round of review and you feel that this manuscript is now acceptable for publication, you may indicate that here to bypass the “Comments to the Author” section, enter your conflict of interest statement in the “Confidential to Editor” section, and submit your "Accept" recommendation.

Reviewer #1: All comments have been addressed

2. Is the manuscript technically sound, and do the data support the conclusions?

Reviewer #1: Partly

3. Has the statistical analysis been performed appropriately and rigorously? 

Reviewer #1: Yes

4. Have the authors made all data underlying the findings in their manuscript fully available?

Reviewer #1: Yes

5. Is the manuscript presented in an intelligible fashion and written in standard English?

Reviewer #1: Yes

6. Review Comments to the Author

Reviewer #1: The authors have adequately addressed most of comments. I have not additional comments on the revised manuscript.

7. PLOS authors have the option to publish the peer review history of their article (what does this mean?). If published, this will include your full peer review and any attached files.

Reviewer #1: No

---

## [Author Response · Author response to Decision Letter 1]

7 Oct 2020

Point by point response is attached with rebuttal letter.

---

## [Editor Report · Decision Letter 2]

21 Oct 2020

PONE-D-20-05351R2

Low contraceptive use, perceived social norms and belief in contraceptive myths among young married women in Eastern Ethiopia

PLOS ONE

Dear Dr. Dingeta,

Thank you for submitting your manuscript to PLOS ONE. After careful consideration, we feel that it has merit but does not fully meet PLOS ONE’s publication criteria as it currently stands. Therefore, we invite you to submit a revised version of the manuscript that addresses the points raised during the review process.

Some of the issues raised have been addressed but others have not. In particular, two points remain:

3- Possible limitation, differences with DHS protocol.

You argue that this is not an issue, but I cannot judge this based on the information I have. I am concerned, for instance, about probing questions regarding knowledge that can make a difference (and are known to make a difference). Please provide a copy of the questionnaire so that this can be assessed. If the protocol is exactly the same, please make it clear in the paper If it is different, acknowledge that as a possible limitation and describe in what ways it is different.

7- I insist in the model with an ever-mother covariate. You could include both models for comparison, but I think it is even more important after showing the strong bivariate link, As I said there are important reasons to include it: married youth with no children are very likely to desire getting pregnant and, therefore, not to use contraception.

We look forward to receiving your revised manuscript.

Kind regards,

José Antonio Ortega, Ph.D.

Academic Editor

PLOS ONE

---

## [Author Response · Author response to Decision Letter 2]

30 Oct 2020

3- Possible limitation, differences with DHS protocol.

You argue that this is not an issue, but I cannot judge this based on the information I have. I am concerned, for instance, about probing questions regarding knowledge that can make a difference (and are known to make a difference). Please provide a copy of the questionnaire so that this can be assessed. If the protocol is exactly the same, please make it clear in the paper If it is different, acknowledge that as a possible limitation and describe in what ways it is different.

Response: We understand your concern and describe the similarity of our questions with the DHS questionnaire in the revised manuscript. A copy of the DHS questionnaire is found on the following website: http://dhsprogram.com/pubs/pdf/FR328/FR328.pdf . The questionnaire specific to contraception is indicated on ‘APPENDIX E’, under the title ‘SECTION 3. CONTRACEPTION’ and page 402 or pdf page 435 

7-I insist in the model with an ever-mother covariate. You could include both models for comparison, but I think it is even more important after showing the strong bivariate link, As I said there are important reasons to include it: married youth with no children are very likely to desire getting pregnant and, therefore, not to use contraception.

Response: We appreciate your concern and addressed it by including both models (age and ever-mother status) based on your suggestion.

---

## [Editor Report · Decision Letter 3]

11 Nov 2020

PONE-D-20-05351R3

Low contraceptive use, perceived social norms and belief in contraceptive myths among young married women in Eastern Ethiopia

PLOS ONE

Dear Dr. Dingeta,

Thank you for submitting your manuscript to PLOS ONE. After careful consideration, we feel that it has merit but does not fully meet PLOS ONE’s publication criteria as it currently stands. Therefore, we invite you to submit a revised version of the manuscript that addresses the points raised during the review process.

There were two issues left:

- ISSUE 1: 3- Possible limitation, differences with DHS protocol.

You argue that this is not an issue, but I cannot judge this based

on the information I have. I am concerned, for instance, about

probing questions regarding knowledge that can make a

difference (and are known to make a difference). Please provide

a copy of the questionnaire so that this can be assessed. If the

protocol is exactly the same, please make it clear in the paper If

it is different, acknowledge that as a possible limitation and

describe in what ways it is different.

RESPONSE: You are giving me a link to the DHS questionnaire. I know that one. What I was asking was to provide a copy of YOUR questionnaire. I am still unable to assess what I was mentioning. And, as I said, it is not a question of simmilarity. If it is not identical in the protocol (example: probing with knowledge of methods, ...) you have to acknowledge that as a possible limitation and describe in what ways it is different. That is still missing.

ISSUE 2: I insist in the model with an ever-mother covariate. You could

include both models for comparison, but I think it is even more

important after showing the strong bivariate link, As I said there

are important reasons to include it: married youth with no

children are very likely to desire getting pregnant and, therefore,

not to use contraception.

RESPONSE: We appreciate your concern and

addressed it by including both models

(age and ever-mother status) based on

your suggestion.

I welcome the addition of the model. You must see how important is the coefficient on ever-mother. But, still, the model is not commented in the text. You should. Please go back to my original comments on the reason for my insistence with including the variable:

I understand your argument of multicollinearity of age and parity,

also raised by referee 1, but for the reason given above (pressure

to become mother), it would be good to include at least a variable

distinguishing ever-mothers. One would expect the variables of

age lose importance after including it, but it is ok. Also much

lower CP for non-mothers would be expected. Please include the

variable in the multivariate analysis since non-mothers are less

likely to use contraception. Note that l.205 says you are

controlling by parity which seems not to be the case (after the

addition of the variable, it will).

Please comment on this in the main text. Please also change the names of the models that do not read well.  You can call them model 2a and model 2b or models 2 and 3 or something like that.  Please also add some measure of model fit for models 1, 2a and 2b that could be used to choose among them, such as AIC, BIC, Nagelkerke pseudo R-squared, etc. Make sure that the results reported throughout the text refer to the best model and make it clear in the text.

We look forward to receiving your revised manuscript.

Kind regards,

José Antonio Ortega, Ph.D.

Academic Editor

PLOS ONE

---

## [Author Response · Author response to Decision Letter 3]

17 Nov 2020

The detail point by point response is uploaded with the revised manuscript.

---

## [Editor Report · Decision Letter 4]

20 Nov 2020

PONE-D-20-05351R4

Low contraceptive utilization among young married women is associated with perceived social norms and believing in contraceptive myths in rural Ethiopia

PLOS ONE

Dear Dr. Dingeta,

Thank you for submitting your manuscript to PLOS ONE. After careful consideration, we feel that it has merit but does not fully meet PLOS ONE’s publication criteria as it currently stands. Therefore, we invite you to submit a revised version of the manuscript that addresses the points raised during the review process.

There were two issues left.

1. Regarding the questionnaire, you convinced me that you were following the same DHS protocol.  I suggest changing the current sentence "We followed the DHS questionnaire protocol in adapting the questions used to assess contraceptive knowledge and utilization" with something more to the point "We followed the DHS questionnaire protocol for questions on contraceptive knowledge and utilization"

2. I think my instructions were clear "Please also add some measure of model fit for models 1, 2a and 2b that could be used to choose among them, such as AIC, BIC, Nagelkerke pseudo Rsquared, etc. Make sure that the results reported throughout the text refer to the best model and make it clear in the text.". 

What you have done is to include a sentence that reads confusing "We used Akaike’s information criterion (AIC), Bayesian information criterion (BIC), and Pseudo R2 in building the regression models. The model with low AIC and BIC, and high Pseudo R2 scores were considered as the final model. Additionally, the model fitness tested using the Hosmer–Lemeshow goodness-of-fit tests". There is no other mention of AIC, BIC or Pseudo R2 in the text. What is going on?

I will be more clear:

1. Choose ONE model selection criterium. I'd suggest AIC.

2.. Report in table 2 the AICs for the different models. In particular look at what model is best according to the criterium: model 2a or model 2b, and make notice of it in the main text.

3. Make sure that all references in the text and abstract to results from the models refer to the best model from step 2.

COMEBACK ISSUE: The title changed again. It is OK except for "believing" that should be BELIEF.

NEW ISSUE: Since you are highlighting in the paper the connection with the variables on social pressure and contraceptive myths,  I see that in the "limitations" paragraph (399-416) there is no mention regarding possible alternative interpretations of the contraceptive myths variable which are clearly there in an observational setting such as post-hoc rationalization. In particular you should add something along these lines "Regarding the impact of belief in contraceptive myths, it cannot be ruled out that the relationship is not causal but due to post-hoc rationalization of a refusal to use contraception".

We look forward to receiving your revised manuscript.

Kind regards,

José Antonio Ortega, Ph.D.

Academic Editor

PLOS ONE

---

## [Author Response · Author response to Decision Letter 4]

25 Dec 2020

The response to reviewers comment is uploaded

---

## [Editor Report · Decision Letter 5]

4 Jan 2021

PONE-D-20-05351R5

Low contraceptive utilization among young married women is associated with perceived social norms and belief in contraceptive myths in rural Ethiopia

PLOS ONE

Dear Dr. Dingeta,

Thank you for submitting your manuscript to PLOS ONE. After careful consideration, we feel that it has merit but does not fully meet PLOS ONE’s publication criteria as it currently stands. Therefore, we invite you to submit a revised version of the manuscript that addresses the points raised during the review process.

Congratulations on a much improved manuscript. The vast majority of issues regarding substance have been addressed. There are important problems regarding readability, though. This is important since a poorly written manuscript will be much less influential.

A first problem of factual incorrection that needs to be rewritten is this:

Previous studies have reported the highest levels of unmet need, and the lowest and interrupted
contraceptive utilization among young married women in developing countries [8].

The cited study does not say what is being said here. It is not a comparative study, it only studies unmet need of young women. In addition, it is not true that young married women have the highest unmet need levels. For instance, https://www.demographic-research.org/volumes/vol38/45/38-45.pdf show that sexually active unmarried young-women have higher levels of demand and unmet need in most DHS surveys than their married counterparts. If we look at unmet need by age in Ethiopia, for instance DHS 2016, in table 7.10.1 we find that unmet need is highest for married women 35-39, and in table 7.10.2 the same applies to all sexually active women. Table 7.10.2 indicates that 15-19 and 20-24 actually have the lowest levels of unmet need.

Regarding editing, I provide a detailed assessment of problems. In particular the section on variables and measurement is written very poorly. Throughout the text there remain problems of grammatically incorrect sentences. I give you a detailed list of problematic sentences and proposed solutions, but the final text must be checked throughout afterwards. I also insert a couple of problems that are not only grammatical.

associated with women’s contraceptive utilization [15-18]. Studies have also been reported the

85 associated with women’s contraceptive utilization [15-18]. Studies have also reported the

89 Recently, literature increasingly recognizing the strong influence of perceived social norms and

89 Recently, literature increasingly recognizes the strong influence of perceived social norms and

91 perceived social norm is commonly shared beliefs about most people in their social networks’ (reference group) approval of contraceptive practice 92 (injunctive norm), and believe that most

93 people in their network contraceptive behavior (descriptive norm)

91 perceived social norm is commonly shared beliefs regarding people in their social networks’ (reference group) approving of contraceptive practice 92 (injunctive norm), or believing that most

93 people in their network contraceptive behavior (descriptive norm)

parents, and husbands' approval influences married young women’s contraceptive utilization
parents, and husbands' approval influences young married women’s contraceptive utilization
methods [22, 27, 28]. The perception of contraceptives behavior of Friends
methods [22, 27, 28]. The perception of contraceptive behavior of friends
friends using contraceptives
friends are using contraceptives

Moreover,

studies have been shown the influence of belief in myths and misperception or beliefs of rumors
not supported by evidence about contraceptive methods on young married women’s contraceptive
use [23, 30-35].133 questions used to measure the exposure variables and a large number of variables assessed like

Moreover,

studies have been shown the influence on young married women’s contraceptive use of belief in myths and misperception or beliefs in rumors
not supported by evidence regarding contraceptive methods  [23, 30-35].

In a community-based survey in which multiple Likert scale

questions used to measure the exposure variables and a large number of variables assessed like
this study, having the largest possible sample size

In a community-based survey such as this one, where multiple Likert scale

questions are used to measure the exposure variables and there are a large number of covariates, having the largest possible sample size …
error within

135 error even within

147 women who were residing in a similar setting outside of the study area. The female data collectors

147 women who were residing in a similar setting outside of the study area. Female data collectors

participants and who consented
participants who consented
The outcome variable was the current contraceptive utilization.
The outcome variable was current contraceptive utilization.
their spouse were currently using any contraceptive method to prevent or delay pregnancy. Those
responded ‘yes’
their spouse were currently using any contraceptive method to prevent or delay pregnancy. Those
who responded ‘yes’
in-law believe I should use contraceptive”.
in-law believe I should use contraceptives”.
strongly agree (5) and the index score was constructed by summing the items score and dividing
the total scores for the total number of items 179 strongly agree (5) and the index score was constructed by dividing
the total score by the total number of items
the total scores for the total number of items (S2 Table). The possible score range from 1-5 with

180 the total scores for the total number of items (S2 Table). The possible score ranges from 1-5 with

181 a higher score indicates that participants perceive their referents are more approve of their

181 a higher score indicating that participants perceive their referents approve more their

practice(descriptive norm) was assessed by single item
practice(descriptive norm) was assessed by a single item
using contraceptive method
using contraceptive methods
score indicates the stronger the likelihood that respondents thought their peers were using the
contraceptive method.
score indicates the stronger the likelihood that respondents thought their peers were using
contraceptive methods.
The statement include
The statements include
by summing the item scores and divided the total score for the number of ítems
by summing the item scores and dividing the total score by the number of items
scores range from 1-5,
score is in the range from 1 to 5,
To control for possible confounding covariates including contraceptive self‐ efficacy, knowledge**,**
inter-spousal communication, household decision-making autonomy, recent exposure to FP
information and socio-demographic variables such age, educational status, parity (ever206

mother/never-mother), desired number of children and wealth status were included the

207 multivariate analysis model.

The multivariate analysis controls for possible confounding including as covariates contraceptive self‐ efficacy, knowledge**,**
inter-spousal communication, household decision-making autonomy, recent exposure to FP
information and socio-demographic variables such age, educational status, parity (ever206

mother/never-mother), desired number of children and wealth status.

Contraceptive self‐ efficacy was measured by four items asked
Contraceptive self‐ efficacy was measured by four items asking
how confident or sure they could use contraceptive and insist husband’s
how confident or sure they were they could use contraceptive and insist the husband
items include how sure that the woman : could tell her your husband that wanted use of family
planning, … could use family planning, … could discuss topic of family planning with husband,
and … could use family planning even if her husband did not want. The items responses
items include how sure was the woman that : she could tell her husband that she wanted to use family
planning, … she could use family planning, … she could discuss topics of family planning with her husband,
and … she could use family planning even if her husband did not want. The item responses
by the number of item[45]. The scale score range from 1-5
by the number of items[45]. The scale score ranges from 1-5
level of confidence to use contraceptive methods. The scale had good reliability scale (Cronbach's
level of confidence to use contraceptive methods. The scale had good reliability (Cronbach's
Spousal communication was measured using a 4-items to ask the respondent how often woman
Spousal communication was measured using 4-items asking the respondent how often the woman
may discuss things with her husband [45]. Each woman was asked how often she discuss with

221 may discuss things with her husband [45]. Each woman was asked how often she discusses with

223 have, and whether to use contraceptive methods. The item response option ranges

223 have, and whether to use contraceptive methods. The item response options range

a range of key household decisions that affect her life and measured using the 10-items scale
a range of key household decisions affecting her life and measured using the 10-items scale
Contraceptive knowledge was another covariate assessed using 15 items adapted from literature
and emphasized on knowledge of specific contraceptive methods
Contraceptive knowledge was another covariate assessed using 15 items adapted from the literature
and emphasizing knowledge of specific contraceptive methods
obtained by summing all items. The score range from

240 obtained by summing all items. The score ranges from

245 whether they were told about FP during their visit to health facility, to Health Extension Worker

(HEW) or others Health care Workers (HCWs) who visited their home

245 whether they were told about FP during their visit to a health facility, or by a Health Extension Worker

(HEW) or other Health care Workers (HCWs) who visited their home
Wealth index was computed by employing principal component analysis (PCA) using items
assessed household facilities and possessions.
Wealth index was computed by employing principal component analysis (PCA) based on items
assessing household facilities and possessions.
each component. The score of the first component or factor which comprised of several heavily
loaded variables and accounted for the largest variation in the data was categorized into quintiles
where each individual fall into

254 each component. The score of the first component or factor comprising several heavily

loaded variables and accounting for the largest variation in the data was categorized into quintiles
where each individual falls into
Additionally, socio-demographic characteristics including current age, age of marriage
educational status, residence of living (rural and urban), alive and ideal number of children were
included in the analysis.
Additionally, socio-demographic characteristics included in the analysis are current age, age at marriage,
educational status, place of residence (rural and urban), alive and ideal number of children
to the multicollinearity
to multicollinearity
models. The AIC values were compared in successive models, and the lowest value was considered
the best-fit model. The value of AIC showed subsequent reduction throughout the model building
which indicates each model improvement over the previous, illustrating the goodness of fit of the
final regression model (Table 2).
models. The model with the lowest AIC value is model 2b, including parity and excluding age.

(Table 2).

NOTE. They are non-nested models. It is not the subsequent reduction: it is one or the other.

**Abbreviations: **ANC, antenatal car;
**Abbreviations: **ANC, antenatal care
24.6%). Among the study participants, 369 (14.2%, 95% CI: 12.8%-15.5%) were currently using
24.6%). Among the study participants, 369 (14.2%, 95% CI: 12.8%-15.5%) were currently using a
currently using contraceptive methods (p-value <0.001). More than two-third
currently using contraceptive methods (p-value <0.001). More than two-thirds
**Fig 1. This is current contraceptive utilization**
**Fig 1. Current contraceptive utilization**;
utilization **(**AOR= 0.60; 95% CI: 0.49-0.73). Moreover, odd of contraceptive utilization was

315 utilization **(**AOR= 0.60; 95% CI: 0.49-0.73). Moreover, odds of contraceptive utilization were

318 (AOR=9.68; 95% CI: 4.47-20.90), and attended secondary education

318 (AOR=9.68; 95% CI: 4.47-20.90), and who attended secondary education

associated young women contraceptive utilization. Moreover, the more young women belief in in
contraceptive myths was negatively associated with their contraceptive utilization.
associated young women contraceptive utilization. Moreover, the more young women believe in
contraceptive myths was negatively associated with their contraceptive utilization.
DHS reports in Ethiopia [49, 50]. Moreover, our finding is far below 40.7% of Oromia; the region
where this study conducted but higher than3.4% of neighboring Somali regions’ level of utilization
DHS reports in Ethiopia [49, 50]. Moreover, our finding is far below 40.7% of Oromia, the region
where this study conducted, but higher than3.4% of neighboring Somali regions’ level of utilization
close cultural and religious similarities with the Somali region’s population, the region where the
contraceptive prevalence among the lowest in Ethiopia.
close cultural and religious similarities with the Somali region’s population, a region with a
contraceptive prevalence among the lowest in Ethiopia.
income countries [35, 51]. For instance, only 2% of contraceptive users were ever-mother 346 income countries [35, 51]. For instance, only 2% of contraceptive users were not-mothers (I ASSUME THIS IS AN ERROR)
likelihood of contraceptive
likelihood of contraception
marriage to delay the age of marriage
marriage to delay the age at marriage
mainly from husbands and mother in-law. Furthermore, evidence also showed that young women’s
mainly from husbands and mother in-law. Furthermore, evidence also showed young women’s
low contraceptive use in our study could be due to many of them comply with the social
low contraceptive use in our study could be due to many of them complying with the social
expectation which encourages childbearing and disapprove of contraception among young women.
expectation encouraging childbearing and disapproving of contraception among young women.
finding suggested that women can learn or receive support from friendships regarding
finding suggested that women can learn or receive support from friends regarding
myths showed the lower the likelihood of reporting contraceptive use
myths showed the lower likelihood of reporting contraceptive use
previous evidence that identified the negative effect contraceptive related misconceptions on
previous evidence that identified a negative effect of contraceptive related misconceptions on
approval of desired behavior (i.e., perceived social approval), beliefs of what would happen if fail
approval of desired behavior (i.e., perceived social approval), beliefs of what would happen if failing
of the social norm, understand the relations between the various constellations, and design tailored
of the social norm, understanding the relations between the various constellations, and designing tailored
myths and misconceptions about the modern contraceptive method.
myths and misconceptions regarding modern contraceptive methods

CHECK REFERENCES. Some include obvious errors:

2. CSA and ICF, Ethiopia Demographic and Health Sruvey 2016. 2017, Ethiopiam Central
Statistical Agency and ICF International: Addis Ababa, Ethiopia.
6. Guttmacher, I. Adding it up:Investing in Contraception and Maternal and Newborn Health for
Adolescents in Ethiopia, 2018, Fact Sheet, New York: Guttmacher Institute, 2017,
https://www.guttmacher.org/fact-sheet/adding-it-up-contraception-mnh-2017. 2018.
14. Yadav, D. and P. Dhillon, Assessing the impact of family planning advice on unmet need
and contraceptive use among currently married women in Uttar Pradesh, India. PloS one,
2015. 10(3): p. e0118584-e0118584.
14. Yadav, D. and P. Dhillon, Assessing the impact of family planning advice on unmet need
and contraceptive use among currently married women in Uttar Pradesh, India. PLOS ONE,
2015. 10(3): p. e0118584-e0118584.
24. UNICEF, What are Social Norms? How are They Measured? UNICEF

507 / UCSD Center on Global Justice

Project Cooperation Agreement, WORKING PAPER, M. Gerry and M. Francesca,
Editors. 2014 Sandiago, USA

(No dates???)

25. UKaid, Social norms, gender norms and adolescent girls: a brief guide. 2015.
27. Mardi, A., et al., Factors influencing the use of contraceptives through the lens of teenage
women: a qualitative study in Iran. 2018. 18(1): p. 202

JOURNAL, PP?

28. Kate, P. and J. Nicola Social norms stop Ethiopian girls from making safe choices about
pregnancy(unpublished). 2020.

LINK

F.N. and A.R. Veloso. Factors that influence the use of birth control by
Brazilian adolescents. in BBR Conference. 2012. Paulo – SP – Brazil.
35. Mutumba, M., E. Wekesa, and R. Stephenson, Community influences on modern
contraceptive use among young women in low and middle-income countries: a cross-sectional
multi-country analysis. BMC Public Health, 2018. 18(1): p. 430.

PAGES

56. Beyeza-Kashesya, J., et al., "Not a Boy, Not a Child": A Qualitative Study on Young
People's Views on Childbearing in Uganda. African Journal of Reproductive Health / La
Revue Africaine de la Santé Reproductive, 2010. 14(1): p. 71-81.
**S1 Table. Item used for**

594 **S1 Table. Items used for**

598 **S1 Minimal data set**

598 **S3 Minimal data set**

We look forward to receiving your revised manuscript.

Kind regards,

José Antonio Ortega, Ph.D.

Academic Editor

PLOS ONE

---

## [Author Response · Author response to Decision Letter 5]

20 Jan 2021

All the comments are addressed in the revised manuscript

---

## [Editor Report · Decision Letter 6]

28 Jan 2021

PONE-D-20-05351R6

Low contraceptive utilization among young married women is associated with perceived social norms and belief in contraceptive myths in rural Ethiopia

PLOS ONE

Dear Dr. Dingeta,

Thank you for submitting your manuscript to PLOS ONE. After careful consideration, we feel that it has merit but does not fully meet PLOS ONE’s publication criteria as it currently stands. Therefore, we invite you to submit a revised version of the manuscript that addresses the points raised during the review process.

There are still problems of readability. In addition, you should have given a point by point response to the comments. Otherwise it makes revision very difficult. There are still many sentences that are incomplete or with no agreement in number among nouns and verbs, or missing (or with superfluous) pronouns. After a reread I include some, but this is still a problem. These are some of the issues remaining but you are encouraged to copy-edit your article further.

Despite the increasingly wider availability of contraception and the level of needs

33 in rural Ethiopia,

I asume you mean “Despite the increasingly wider availability of contraceptives and the high levels of potential demand [or unmet need] for family planning in rural Ethiopia, …”

Substantial proportion of young married women in low and middle-income countries are at risk of

73 unintended pregnancy due to low contraceptive utilization than their adult counterparts. Young

74 women in the region also often used short-term methods that related to high failure and

75 discontinuation (8, 9).

The sentence is not grammatically or factually correct: what does the “than” mean, what is the comparison? As I said in the previous revision, what matters here is unmet need, and the levels are not highest among young married women. Also, women over 18 are adults. As it was discussed it is not true that levels of unmet need are highest. The second sentence is also missing a verb (I guess *are* related to higher ..).

Change proposed to old line 89-93 only partially adopted (missing the second part), and it now reads funny.

New line 102: studies have shown, not “studies have been shown”

New line 203: higher score indicates high believe in the myths or had 204 misconception about contraceptive methods.

Should be “a higher score indicates strong belief in the myths or  misconceptions regarding contraceptive methods.”

216 or sure they were they could use contraceptive and

216 or sure they were they could use contraceptives and

221 … she could use family planning even if her husband did not want. The items responses

221 … she could use family planning even if her husband did not want. The item responses

224 by the number of item (

224 by the number of items (

231 The item response option

The item response options

252 Exposure to FP information in last 12 months

252 Exposure to FP information in the last 12 months

utilization **(**AOR= 0.60; 95% CI: 0.49-0.73). Moreover, odd of contraceptive utilization
utilization **(**AOR= 0.60; 95% CI: 0.49-0.73). Moreover, odds of contraceptive utilization

In several places, including table 1 and mention of variables: replace “Number of alive children” with “Number of children alive”

Also in this context “Had no children” replace with “Has no children” or, even better “No children alive”.

Also in this context, replace “Three and above children” with “Three and more children”

where this study conducted, but higher than 3.4% of neighboring Somali regions’ level of
utilization among reproductive-age women(2).
where this study was conducted, but higher than 3.4% in the neighboring Somali region (2).

396 with current contraceptive utilization. The more young women belief in contraceptive related

397 myths showed the lower the likelihood of reporting contraceptive use.

396 with current contraceptive utilization. The more young women believe in contraceptive related

397 myths, the lower the likelihood of reporting contraceptive use.

previous evidence that identified a negative effect contraceptive related
previous evidence that identified a negative effect of contraceptive related

434 myths and misconceptions regarding the modern contraceptive methods

434 myths and misconceptions regarding modern contraceptive methods

We look forward to receiving your revised manuscript.

Kind regards,

José Antonio Ortega, Ph.D.

Academic Editor

PLOS ONE

---

## [Author Response · Author response to Decision Letter 6]

4 Feb 2021

point by point response is attached

---

## [Editor Report · Decision Letter 7]

5 Feb 2021

Low contraceptive utilization among young married women is associated with perceived social norms and belief in contraceptive myths in rural Ethiopia

PONE-D-20-05351R7

Dear Dr. Dingeta,

We’re pleased to inform you that your manuscript has been judged scientifically suitable for publication and will be formally accepted for publication once it meets all outstanding technical requirements.

There is one small change that still needs to be incorporated: In line 92 "or believing that most people in their network contraceptive behavior" is missing a verb. It should be "are using contraceptives" or "are practicing contraception".

Kind regards,

José Antonio Ortega, Ph.D.

Academic Editor

PLOS ONE
---

## [Editor Report · Acceptance letter]

11 Feb 2021

PONE-D-20-05351R7 

Low contraceptive utilization among young married women is associated with perceived social norms and belief in contraceptive myths in rural Ethiopia 

Dear Dr. Dingeta:

I'm pleased to inform you that your manuscript has been deemed suitable for publication in PLOS ONE. Congratulations! Your manuscript is now with our production department. 

Kind regards, 

on behalf of

Dr. José Antonio Ortega 

Academic Editor

PLOS ONE